# Optimization of Plant Nutrition in Aquaponics: The Impact of *Trichoderma harzianum* and *Bacillus mojavensis* on Lettuce and Basil Yield and Mineral Status

**DOI:** 10.3390/plants13020291

**Published:** 2024-01-18

**Authors:** Kateřina Patloková, Robert Pokluda

**Affiliations:** Department of Vegetable Sciences and Floriculture, Mendel University in Brno, 69144 Lednice, Czech Republic; robert.pokluda@mendelu.cz

**Keywords:** PGPMs, microorganisms, bacteria, fungi, inoculation, *Lactuca sativa*, *Ocimum basilicum*

## Abstract

The present study aims to test the effect of a nutrient solution, with the addition of microbial inoculum, on the growth and mineral composition of ‘Hilbert’ and ‘Barlach’ lettuce cultivars (*Lactuca sativa* var. *crispa*, L.) and basil (*Ocimum basilicum*, L.) cultivated in a vertical indoor farm. These crops were grown in four different variants of nutrient solution: (1) hydroponic; (2) aquaponic, derived from a recirculating aquaculture system (RAS) with rainbow trout; (3) aquaponic, treated with *Trichoderma harzianum*; (4) aquaponic, treated with *Bacillus mojavensis*. The benefits of *T. harzianum* inoculation were most evident in basil, where a significantly higher number of leaves (by 44.9%), a higher nitrate content (by 36.4%), and increased vitamin C (by 126.0%) were found when compared to the aquaponic variant. Inoculation with *T. harzianum* can be recommended for growing basil in N-limited conditions. *B. mojavensis* caused a higher degree of removal of Na^+^ and Cl^−^ from the nutrient solution (243.1% and 254.4% higher, in comparison to the aquaponic solution). This is desirable in aquaponics as these ions may accumulate in the system solution. *B. mojavensis* further increased the number of leaves in all crops (by 44.9–82.9%) and the content of vitamin C in basil and ‘Hilbert’ lettuce (by 168.3 and 45.0%) compared to the aquaponic solution. The inoculums of both microbial species used did not significantly affect the crop yield or the activity of the biofilter. The nutrient levels in RAS-based nutrient solutions are mostly suboptimal or in a form that is unavailable to the plants; thus, their utilization must be maximized. These findings can help to reduce the required level of supplemental mineral fertilizers in aquaponics.

## 1. Introduction

Plant production, as part of aquaponics, currently faces a number of challenges. Firstly, there are suboptimal nutrient levels (mainly reduced K^+^, Mg^2+^, Ca^2+^, total Mn, Fe and sometimes even PO_4_^3−^) and ratios [1,2,3]. Secondly, there is a lack of studies into the transport of nutrients through the subunits of the system (in the past, attention was paid to the N balance in particular). Finally, it is necessary to determinate the efficiency of nutrient use for elements other than N [2,4]. Suboptimal nutrient levels can be overcome in several ways, such as working in the so-called decoupled aquaponic system (DAS), which allows the addition of mineral fertilizers [5]; alternatively, it is possible to apply foliar fertilizers [6]. However, aquaponics research is also focused on other, more sustainable solutions that will make it possible to use the nutrients that are already present in the system to the maximum possible extent [2]. One possibility is to optimize the quantity and appropriate type of fish feed or design new formulations of fish feed with regard to the further use of nutrients released into solution in the hydroponic subsystem [7]. Another very promising area is the use of so-called plant-growth-promoting microorganisms (PGPMs) [4]. The group of PGPMs, according to Debasis et al. [8], are made up of rhizo-bacteria, actinomycetes, fungi (incl. the group of arbuscular mycorrhizal fungi), and endophytes, which occur in the rhizosphere and promote plant growth via various mechanisms. Whichever method is chosen to adjust the nutrient solution, the type of crop under cultivation and the developmental stage of the plant must also be considered [9].

Genus *Trichoderma* is part of the *Hypocreaceae* family and is mainly known for its biocontrol properties—mycoparasitism and the induction of plant defence [10,11]. However, in addition, there is evidence of its decomposition, bioremediation, and detoxification effects [12,13,14], as well as its promotion of plant growth and germination [15]. A large amount of data, that demonstrate its positive effect on the promotion of plant growth after *Trichoderma* spp. inoculation, have been summarised in the literature (e.g., by Abdul-Halim et al. [16]). The key mechanism of this phenomenon is that *Trichoderma* spp. produces phyto-hormones and phyto-reagents such as IAA (indole-3-acetic acid), GA_3_ (gibberellic acid), ACCD (ACC-deaminase enzyme), and harzianolid, which stimulate root growth and development. As a result, *Trichoderma* increases the available root absorption surface while simultaneously promoting nutrient intake [15,17,18]. Vinale et al. [19] found that *Trichoderma* forms harzianic acid, a secondary metabolite which has a good affinity to essential metals such as Fe^3+^, which may alter the uptake of this microelement by the plant. The inoculation of cucumber plants with the *Trichoderma harzianum* T-203 strain was carried out in experimental conditions by Yedida et al. [20]. Part of the experiment was carried out in soil conditions, where a significant increase in dry weight, shoot length, and leaf area were observed along with an increase in P and Fe (by 90 and 30%) in plant tissues. In the hydroponic section of the experiment, an increase in the amounts of Zn, P, and Mn (by 25, 30, and 75%, respectively) were found in the plant tissues.

The effective role of plant-growth-promoting rhizobacteria (PGPR) in enhancing the plant growth, yield, and quality has been reported [21]. Otherwise, the production of residual-free products from edible crops has gained more attention, guaranteeing their quality and safety [22]. Members of the *Bacillus* genus are certainly some of the most important PGPRs. These ubiquitous, rod-shaped, Gram-positive aerobic bacteria have the ability to form spores [23]. Their ability to promote plant growth lies in a variety of regulatory mechanisms and microbe–host signalling. Tsotetsi et al. [23] summarized the main mechanisms of action as follows: (i) siderophore production (e.g., enterobactin, pyochelin, alcaligin and rhizoferin) and upregulation of Fe acquisition genes by *Bacillus* spp. facilitates Fe uptake by the plant; (ii) solubilization of nutrients such as N or P through the production of lytic enzymes and low-molecular-weight acids (such as formic, acetic, or lactic acid); (iii) production of phytohormones (IAA, cytokinins, gibberellins, or abscisic acid) and volatile organic compounds (VOCs) which are directly linked to the modification of root architecture and improved plant growth; (iv) biofilm formation and the production of extra polymeric substances (EPS), NH_3_ and HCN, are indirectly linked to plant growth promotion via crop protection against abiotic stresses [23,24].

It has been demonstrated that, within the soil environment, members of the PGPMs group, for instance, enhance nutrient absorption by plants (and thus also increase activation of the primary and secondary metabolic cycles) and consequently increase the yield of these plants [25,26,27]. Furthermore, PGPMs cause a change in root morphology and mitigate the effects of stress caused by drought, salinity, or low temperature [28,29,30,31]. In addition, the protective effect of these microorganisms in relation to soil pathogens was demonstrated [32]. Although research on PGPMs in a soil-free environment currently lags behind that in soil, according to Barteleme et al. [4], certain parallels among plant–microorganism interactions from a soil environment are preserved. This statement is supported by studies from Gravel et al. [33], Cerozi and Fitzsimmons [34], and Sheridan et al. [35] or review articles by Dhawi [36] or Mourouzidou et al. [37].

It was hypothesized that the selected species of PGPMs (*Trichoderma harzianum* and *Bacillus mojavensis*) (1) can survive and interact with plants in the aquatic environment; (2) can increase the uptake of some nutrients from the aquaponic nutrient solution; (3) do not affect the activity of nitrifying bacteria. The objective of the present research was to assess how lettuce and basil, cultivated in nutrient solutions derived from various nutrient sources, differ in terms of growth and nutrient absorption. The primary aim was to investigate whether incorporating specific species of PGPMs influences the utilization of nutrients from the aquaponic nutrient solution and therefore has the potential to solve one of the most crucial challenges in aquaponics.

## 2. Results and Discussion

### 2.1. Plant Nutrient Uptake Experiment

#### 2.1.1. Microscopic Detection

The presence of microorganisms was verified using microscopic methods. *B. mojavensis* detected in root samples can be seen in Figure 1. The different density of microorganisms can be caused by two factors: (a) there is a symbiotic relationship among ‘Barlach’ lettuce variety and *B. mojavensis*, meaning, e.g., that root exudates could support the growth of this microorganism; (b) the root fragment pictured in Figure 1B could have been taken from a part of the substrate that was not submerged in water, thus better oxygenated, which promoted the higher development of *B. mojavensis*.

#### 2.1.2. Biometric Parameters, Dry Weight

The ‘Barlach’ lettuce and basil showed the highest total weight and the weight of above-ground biomass (yield) when grown in the HYDRO variant of the nutrient solution (Table 1). On the contrary, the ‘Hilbert’ lettuce showed comparable results for these parameters in all solutions (Table 1, Figure 2A). This may indicate lower nutritional requirements or sensitivity to nutritional deficiencies of ‘Hilbert’ variety. The addition of the *T. harzianum* or *B. mojavensis* inoculum neither increased nor decreased the total yield of all the studied crops. Although the level of Na^+^ and Cl^−^ was higher in the AQP variants of the solution when compared to the HYDRO variant, there was no growth restriction observed in ‘Hilbert’ lettuce crop, which indicates that ‘Hilbert’ is a suitable variety for aquaponics. This finding is in accordance with the findings of Goddek’s and Vermeulen’s study [38], which demonstrated that lettuce is relatively sodium insensitive. Goddek and Vermeulen [38] found that the variant fed by RAS-derived water had a 7.9% higher fresh weight compared to the hydroponic control. The present study did not confirm an increase in yield. The root weight was comparable for all of the different variants of the solution. Basil grown in the HYDRO solution had a significantly lower ratio of above-ground biomass to root biomass when compared to the AQP variants. The roots of the basil grown in the HYDRO solution only made up 29.6% of the above-ground weight, whereas when grown in AQP, AQP + TricH, and AQP + BM it was 40.9, 39.7, and 36.9%, respectively. The root development of the basil was comparable for all solutions. Figure 2B shows the number of leaves on the crops under examination for all variants of nutrient solution. This parameter was significantly higher for all crops treated with *B. mojavensis* when compared with AQP variant. Basil treated with *T. harzianum* also showed a significantly higher number of leaves when compared to the AQP solution. Although there were no significant differences observed in DW, basil grown in the AQP solutions had a higher average dry matter content in comparison to the HYDRO variant. A similar result was observed by Goddek and Vermeulen (2018), who found that the DW of lettuce grown in RAS-based water was up to 33.2% higher in comparison to the hydroponic variant.

Inoculation with *B. mojavensis* has a positive effect on the number of leaves of all examined crops; the same effect was observed in basil treated with *T. harzianum*. Simultaneously, there was no decrease in yield observed in inoculated variants.

#### 2.1.3. Nutrient Solution Parameters

The usual pH range of hydroponic solutions is between 5.5 and 6.5 [39]. In aquaponics, there are two options, depending on the system design (coupled/decoupled aquaponic system). In a single-loop system (the so-called coupled or CAS), it is common to maintain a neutral pH (a compromise for the fish/bacteria/plants). The other option is the so-called decoupled system or DAS; this allows the operator to adjust the water parameters for plants, so the pH value is reduced to the hydroponic optimum [40]. As can be seen from Table 2, the pH of the hydroponic solution was suitable for hydroponic growth throughout the entire experiment. In the present study, the pH and EC values of water from the RAS with rainbow trout were not adjusted to the optimum for hydroponic growth. The RAS-derived water initially had a neutral pH, and this even increased during the duration of the experiment. In CAS aquaponics, a neutral pH is commonly maintained, and it does not seem to be a limiting factor [2]. However, the pH increased considerably for all the AQP solutions and especially for the inoculated variants (see Table 2 and Table 3). Since the systems were not connected to the biofilter, the alkalization of the AQP solutions probably occurred during the assimilation of nitrates by the plants (the formation of OH^−^ and HCO^3−^) [1] and through microbial activity. The alkalization of the AQP variants compared to the HYDRO solution indicates that, although all solutions were initially exposed to UV-C radiation, the RAS derived microbiota was probably not completely eliminated.

Within the inoculated variants, TAN also increased; in the case of *T. harzianum,* it probably indicates the partial decomposition of organically bound nitrogen from the root area (whereas *Trichoderma* prefers ammonia-N and amino acids as a source of N) [41]. *B. mojavensis* is known for its ability to produce ammonia-N, even when stressed, as reported by Danish et al. [24]. The increase in the level of ammonia in both the inoculated variants resulted in an increase in the pH value. According to Lennard and Goddek [42], in higher pH conditions, a reduction in the uptake of nutrients from the solution can be expected, especially for micronutrients such as iron, manganese, boron, copper, and zinc. The pH of the water coming from the RAS of rainbow trout is similar to that observed in the study by Yang and Kim [43] in Nile tilapia farming. However, on the contrary, it is lower than the value found by Goddek and Vermeulen [38] in common carp RAS-derived water or the values found in three out of four tanks containing Nile tilapia in the research conducted by Sanchez, Vivian-Rogers, and Urakawa [44]. The pH level can be controlled by the aquaponic farm operator depending on which of the production outputs are more valuable on a particular farm [2].

Pokluda and Kobza [45] state that an EC of 1.2–1.6 mS/cm is suitable for growing lettuce; from this point of view, the initial EC of all solutions was optimal. In the HYDRO solution, the EC value dropped by 37.5% during the experiment, which suggests a relatively high uptake of the nutrients NO_3_^−^, TAN, K^+^, and F^−^ (see Table 2); this was probably a result of a better balance of nutrients and an optimal pH value. The EC value of the RAS-derived solutions (AQP variants) is equal to or greater than those found by Sanchez, Vivian-Rogers, and Urakawa [44] or Goddek and Vermeulen [38].

Both of the added inoculants caused alkalization of the nutrient solution by an increase in TAN level. To ensure optimal nutrient utilization, it is essential to assess and adjust the pH value of the nutrient solution when inoculation with *T. harzianum* or *B. mojavensis* is employed in aquaponics.

#### 2.1.4. Nutrient Uptake and Levels in Plants

The uptake of nutrients from the system by plants or microbiota could be monitored based on the addition of deionised water to the original volume during the experiment.

The level of phosphates in the RAS-derived water was significantly lower than optimal, even at the beginning of the experiment (see Table 2). The amounts of phosphates taken up from the solutions were comparable for the AQP and AQP + TricH variants. In contrast, the *B. mojavensis* variant showed a significantly lower level of phosphate uptake when compared to the AQP variant. The lowest level of phosphate uptake was recorded for the HYDRO variant, which was caused by the precipitation of P and Ca into insoluble forms such as calcium phosphate (Ca_3_(PO_4_)_2_) and dicalcium phosphate (CaHPO_4_) [46]; these cannot be used by the plants. This was probably caused by the high calcium content of the well inlet water (140 ± 13 mg/L) and the pH spike above 6.3 during the set-up phase of the experiment. Thus, a comparison of the PO_4_^2−^ and Ca^2+^ in the HYDRO and AQP solutions is not relevant. Although there were no significant differences identified when comparing all AQP solutions, it can be seen from Table 4 that the mean values of P in the AQP + TricH and AQP + BM treatments for basil were higher (by 35.5 and 28.7%, respectively) in comparison to the AQP variant. Despite the lack of phosphates in the nutrient solution of all AQP variants, the ‘Hilbert’ variety showed a comparable P content in all the solution variants. Table 5 shows the nutrient use efficiencies for P, K, Ca, and Mg. This parameter compares the value of the amount of the analyte fixed in the plant tissues with the amount of analyte entering the system. As can be seen from the table, the PUE was significantly higher in the AQP + BM variant compared to the HYDRO variant. The PUE values obtained in the present study differ from the values obtained by Cerozi and Fitzimmons [47]. The PUE of 29.4% which they found refers to the initial value of phosphorus contained in the fish feed. The values stated in Table 5 refer to the initial level in the solution.

A low potassium level was found in the HYDRO solution (in relation with nitrate content), which further deteriorated as nitrates were removed from the solution during the experiment. A comparable situation was observed in the RAS-derived nutrient solutions. After the potassium content was related to the quantity of nitrates, it was calculated that the RAS-derived water only contained 10.7% of the required K. With nitrate withdrawal, this percentage increased slightly during the experiment in all AQP variants. Yep and Zheng [2] state that K limitation in aquaponic nutrient solution has the greatest significance. This agrees with results of this study, where at T_0_ the most significant deficiencies in the AQP solutions were recorded for iron, followed by potassium and phosphorus. The potassium uptake from the AQP solutions was analogous as can be seen in Table 3. The low level of K in the AQP solutions was also reflected in the total amount of K found in the plant tissues (Table 4). However, Table 5 further indicates that in the AQP variants, more than 70% of the input K was incorporated into the plants, which is considerably higher than that observed for the HYDRO solution. This finding is in agreement with Goddek and Vermeulen [38], who also observed a stimulated K uptake from RAS-derived water.

The initial level of Ca in the HYDRO solution was very low due to precipitation. In contrast, the initial value in the AQP nutrient solution was relatively high, close to the optimum. The Ca value detected in the AQP solution derived from RAS with rainbow trout is higher than the Ca detected, for example, in RAS-derived water from common carp [38]. The influence of the inoculum on the removal of Ca from the solution was not demonstrated (Table 3). Analyses of the plant tissues did not show any significant differences between the AQP variants; however, there was an increase in the mean Ca in ‘Barlach’ lettuce inoculated by *T. harzianum* of 137.6% when compared to ‘Barlach’ grown in the AQP solution. However, an insignificant reduction (by 46.0%) was noted in the ‘Hilbert’ lettuce grown in the AQP + TricH solution when compared to the same crop grown in the AQP solution. This finding points to a specific PGPM-host response. According to Saravanakumar, Arasu, and Kathiresan [48], all the *Trichoderma* isolates tested in the study showed in vitro solubilizing activity towards Ca_3_(PO_4_)_2_, indicating an improved availability of both nutrients for plants.

Manifestations of K deficiency could be further accentuated by an excess of Mg in the HYDRO solution, which could reduce total K uptake. As with the other ions in the solution (except NO_3_^−^ and Ca^2+^), magnesium deficiency was also noted in the AQP nutrient solutions. Basil grown in the AQP + TricH and AQP + BM variants had significantly higher amounts of magnesium in their tissues compared to basil grown in the HYDRO solution. A similar trend was not identified for both of the lettuce varieties; this contradicts the finding of Goddek and Vermeulen [38], where a significantly higher amount of Mg was found in lettuce grown in RAS-derived water. All of the aquaponic variants had higher mean levels of MgUE compared to the HYDRO solution, while the variant with *T. harzianum* showed significantly higher levels of MgUE. This process may be caused by the production of organic acids (such as fumaric, citric, or gluconic) by *Trichoderma* spp, which lowers the pH and increases the availability of phosphates, magnesium, iron, or manganese [18].

There was a high concentration of sulphates in the HYDRO solution from the beginning, which could cause a significantly lower removal rate from the solution when compared to the AQP variants. The accumulation of sulphates in the controlled production of plants with recirculation of the nutrient solution is a common problem, this phenomenon increases the EC of the solution and reduces the exploitability of some cations, e.g., calcium [49].

The concentration of Na^+^ in the solutions was calculated based on the Cl^−^ values. The values stated in Table 2 therefore do not include, for example, Na^+^, which originates from sodium bicarbonate (NaHCO_3_) and is commonly used in RAS to neutralize the solution [50]. In RAS, NaCl is also commonly added in order to mitigate stress, for the prevention and control of diseases, and for the adjustment of osmoregulation [38]. A significantly higher uptake of both ions (Na^+^ and Cl^−^) from the solution was found for the AQP + BM variants. This is potentially a method that could prevent the accumulation of these ions in the RAS-derived solution. Goddek and Vermeulen (2018) further suggest the incorporation of halophytes in the hydroponic part of the aquaponic system. The ratio of a definite species of halophytes needs to be specified in the future in terms of the amount of present salt. Another option, which is more expensive, is the inclusion of a desalination unit in the aquaponic system [40]. The fact that the Na^+^ values in the AQP nutrient solutions were higher is also indicated by Figure 3A, which shows the amount of Na incorporated into the plant tissues. Very high values of Na content were recorded for both of the lettuce varieties; by contrast, basil appears to be a highly Na-tolerant plant.

#### 2.1.5. Chlorophyll, Carotenoids, and Vitamin C

There were no significant differences observed in the content of chlorophyll a and b within the crops; however—for example in the case of ‘Barlach’ lettuce that was grown in the AQP + BM solution—the mean content of chlorophyll a and b was higher when compared to that grown in the HYDRO or AQP solutions. Basil grown in the AQP + TricH variant also showed higher mean values of both chlorophyll a and b when compared to the control AQP and HYDRO solutions (see Table 4). Higher chlorophyll content in plants when inoculated with *Trichoderma* was observed by Uddin et al. [51] and Andrzejak and Janowska [41]; contrary to their findings, Vukelić et al. [52] observed a 13% reduction in chlorophyll in plants inoculated with *T. harzianum*. As can be seen from Figure 3B, the highest carotenoid content was found in basil grown in the AQP + TricH solution. A significantly lower carotenoids content (by 46.4%) was found in the HYDRO variant. When compared to AQP, the difference was not significant, yet the mean value of carotenoids was 24.2% higher in the AQP + TricH variant. A higher level of carotenoids in ornamental plants inoculated with *Trichoderma* was also observed by Andrzejak and Janowska [41]. Figure 3C shows the effect of the individual variants of the nutrient solution on the vitamin C content. In general, the mean value for the vitamin C content in all the crops was higher for all the AQP variants when compared to the HYDRO solution. Both *T. harzianum* a *B. mojavensis* significantly increased the level of vitamin C in basil (by 126.0% and 168.3% when compared to AQP and by 876.7% and 1059.7% when compared to HYDRO). A significant increase in vitamin C was also found in the ‘Hilbert’ lettuce grown in the AQP + BM solution (by 45.0% when compared to AQP and by 87.7% when compared to HYDRO) and in the ‘Hilbert’ lettuce when grown in AQP + TricH (by 40.0% when compared to HYDRO). Jamil [53] reported a significantly higher amount of vitamin C in tomato plants when treated with either *Trichoderma viride* or *T. harzianum*. A mixture of *Trichoderma atroviride* LX-7 and *T. citrinoviride* HT-1 in a 1:2 ratio provided the highest values of vitamin C in pakchoi plants [54].

#### 2.1.6. Nitrates

Although the original nitrate content in the nutrient solutions was higher in the AQP variants, the highest total uptake of nitrates, based on the decrease in nutrients in the solutions, was observed for the HYDRO variant (decrease of 40.0%), followed by the AQP + BM variant (33.0%), the AQP (28.2%), and the AQP + TricH (22.9%) (see Table 3 and Table 6). Although there was no significant difference observed between the nitrate uptake of the AQP + BM and AQP variants, the uptake of the inoculated variant was still almost 17% higher. This confirms the findings of Prajakta et al. [55], which showed that *B. mojavensis* exhibits nitrogen fixation activity. Considering the nitrates content in the plant tissues (Figure 3D), there was no increase observed for the AQP + BM variant compared to the AQP variant. The high level of decline of nitrates from the solution could either manifest as incorporation in the plants in the form of amino acids (AA) or proteins or it could have mostly been utilised by the microbial inoculum. The impact of *Bacillus* spp. on the amount of AA and proteins in the plant tissues in aquaponics needs to be examined in the future. The greater degree of removal of nitrates from the AQP + BM variant, found in the present study, could be supplemented by the results of Kasozi, Kaiser, and Wilhelmi [56], who monitored the concentration of nitrates in the aquaponic system water inoculated with *Bacillus* species (*B. subtilis* + *B. licheniformis*). Both trials in their study demonstrated a significantly higher increase in nitrates with an inoculated variant and they attribute this to enhanced biological nitrification caused by the *Bacillus* inoculum. If we put these two conclusions together, the inoculation of the aquaponic system with *Bacillus* spp. may result in a higher nitrate formation and simultaneously increased N uptake by the plants. According to Pandey et al. [57], there was an increase in crude protein of 22.1% in amaranth seeds after inoculation with a mixture of *Bacillus pumilis* and *Bacillus subtilis*, which suggests that a similar effect may have been involved in the present study.

Contrary to that, in the *T. harzianum* variant, a significantly lower uptake of nitrates from solution was observed, but the quantity of nitrates in the basil was significantly higher (by 36.4%) when compared to the AQP variant. Singh et al. [58] give examples of several soil experiments where *Trichoderma* increased nitrogen use efficiency (NUE). The mechanism of this phenomenon is still not fully understood, but in the presence of *Trichoderma*, there is, for example, an increased transcription of nitrate reductase in the plant cells, which leads to the reduction of nitrates to nitrites and subsequently to nitric oxide, which serves as an N-signalling molecule. This induces the production of nitrate transporters (NRTs) that facilitate NO_3_^−^ uptake by the plant [58]. Considering that in the aquaponic cultivation of leafy greens or herbs there is usually no problem with a lack of NO_3_^−^ in the solution [2], the observed phenomenon is rather undesirable, as excessive accumulation of nitrates may occur in basil when treated with *T. harzianum*. Day et al. [59] points out that animal welfare may become stricter in the future; therefore, inoculation with *T. harzianum* may be recommended in aquaponic basil production in N-limited conditions, e.g., when the RAS part of the aquaponics operates in an extensive mode.

The total nitrogen in plant tissues was not examined through an analytical approach in this study. Theoretical values for nitrogen use efficiency were calculated as the output N/input N × 100 (Table 6) [60]. This value assumes that all of the reduction in the level of nitrates in the nutrient solution was incorporated into the plant tissues. After taking the average of the values of N fixed in nitrates vs. total nitrogen in lettuce tissue samples, from the study carried out by Marsic and Osvald [61], where lettuce was grown in a concentration of 13 mM NO_3_^−^/L, the percentage of 1.4% N fixed in nitrates out of total N may be calculated.

### 2.2. Biofilter Experiment

During the experiment, TAN and nitrites (Figure 4A,B) decreased in all variants (Control, TricH, and BM) and followed a similar trend with no significant differences observed. The nitrates (Figure 4C) were initially at a high level, and slightly decreased for all variants. This was probably caused by the time gap between the removal of the samples from the biofilter and the commencement of the experiment, i.e., the addition of NH_4_Cl and reintroduction of aeration. Then, the value in all variants increased. The same trend was recorded for all analytes in both inoculated variants when compared to the non-inoculated control. This indicates that none of the microorganisms used in this study affects the activity of the biofilter at the dosage recommended by the inoculum manufacturer. Nevertheless, it would be necessary to monitor this over a longer period of time in order to see whether there is competition and the possible overgrowth of any of the species of microorganism.

## 3. Materials and Methods

### 3.1. Plant Nutrient Uptake Experiment

#### 3.1.1. System Setup and Cultivation Conditions

The experiment was conducted in an indoor vertical growing system in the Future Farming Ltd. R&D centre (Kaly 66, 49.3792886 N, 16.3515736 E, Czechia). The growing system consisted of four racks, where plants were cultivated through a deep-water culture (DWC) technique. Each rack was connected to its own individual sump tank. Each rack, its pipes, and sump tank represented a single closed system with a volume of approximately 180 L (Figure 5). In total, there were four closed systems (Figure 6), each of which contained one of the four variants of the nutrient solution. The water from each system was constantly circulated; the water pump drove the water up to the rack with the plants and it was returned by gravity to the sump tank. Each rack had LED lighting installed (R:B 2.5:1) at the same height (220 mm) from the cultivation site, which were set up to provide an 18 h/day (6:00–0:00) with a photosynthetic photon flux density (PPFD) of 130 µmol/m^2^/s. The temperature of the growing space was set to 20 °C, which is a compromise between the demands of basil and lettuce. The relative water humidity (RWH) was 65% throughout the experiment. No additional CO_2_ was supplied.

#### 3.1.2. Experimental Design

The experiment was arranged in a completely randomized design (CRD). Plants, as well as nutrient solution variants, were randomly allocated to individual cultivation systems (racks). The experiment comprised of four variants, each of which had one repetition (one cultivation rack). The number of measurement repetitions is specified in individual analyses (tables with results). The experimental setup is shown in Figure 6. There were two variables in the experiment: the nutrient solution and plant type. The variants of nutrient solution were as follows:(1)HYDRO (control)—hydroponic nutrient solution (commercial fertilizer for leafy vegetables, made up of two components: JUNGLE garden G1 0.4% N, 2% P_2_O_5_ and 4.5% K_2_O and JUNGLE garden BASE 7% N, 11.2% CaO and 0.22% Fe, Numazon Ltd., Brno).(2)AQP—aquaponic nutrient solution derived from a recirculation aquaculture system (RAS) with rainbow trout (*Oncorhynchus mykiss*, Walbaum).(3)AQP + TricH—the aquaponic nutrient solution inoculated with *Trichoderma harzianum*.(4)AQP + BM—the aquaponic nutrient solution inoculated with *Bacillus mojavensis*.

Each of the four individual closed systems described above contained a variant of the nutrient solution. The effect of the nutrient solution was tested on 3 vegetable crops: (A) multi-leaf green lettuce ‘Hilbert’ (*Lactuca sativa* var. *crispa*, L., RijkZwaan Ltd., De Lier, The Netherlands); (B) green basil (*Ocimum basilicum*, L., Suba Seeds Company JSC, Longiano, Italy); (C) multi-leaf red lettuce ‘Barlach’ (*Lactuca sativa* var. *crispa*, L., RijkZwaan Ltd., De Lier, The Netherlands). These representatives of leafy vegetables and herbs were chosen for the experiment because these species are commonly and successfully cultivated in aquaponic systems. The experiment included a total of 12 variants and each variant was represented by 10 plants in the system.

#### 3.1.3. Inoculation

Both of the inoculants tested are commercially available products. *Trichoderma harzianum*, Rifai (1969), was delivered to the system as the product TrikoLogic^®^ (Terra Aquatica, Fleurance, France). The product contains 10^8^ spores/g and it was dosed at a ratio of 0.1 g/L of the irrigation water.

*Bacillus mojavensis*, Roberts et al. (1994), was inoculated as the product amazoN–microbial adjuvant (Biovéd 2005 Ltd., Pinkamindszent, Hungary). The product contains *Bacillus mojavensis*, strain KN32, NCAIM 497/2020 with a minimum of 5 × 10^9^ CFU/m^3^, attached to perlite. The inoculum was dosed at a ratio of 1 g/L of the irrigation water.

#### 3.1.4. Course of the Experiment

The experiment began on 24 April 2023 when the seedlings were sown on an inert mineral rockwool substrate (1 cube = 40 × 40 × 40 mm). All the seedlings were pre-grown for three weeks using the hydroponic nutrient solution, while the AQP + TricH and AQP + BM variants were pre-grown in separate rafts to ensure no contamination. These variants were inoculated one week after sowing via irrigation with the hydroponic solution. The conditions for the pre-growth stage were similar to those of the growing stage. After three weeks, the systems were filled with the appropriate nutrient solution. The base of the AQP, AQP + TricH, and AQP + BM variant was a nutrient solution derived from rainbow trout RAS. The clean water input to the RAS was from a local well. InHYDRO variant, a complete hydroponic fertilizer was used, the hydroponic nutrient solution was produced by adding 2.5 mL of JUNGLE garden G1 and 1 mL of JUNGLE garden BASE per 1 L of water from a local well. All the solutions were treated with UV-C light for one hour to eliminate microorganisms and then the second inoculation was performed to produce AQP + TricH and AQP + BM variants. Samples of the nutrient solutions were taken at that time (T_0_). The plants were then transplanted into the four systems (Figure 6), where they were allowed to grow for another four weeks. During that time, all four systems were refilled using deionised UV-C-treated water (180 L of water/1 h of UV-C irradiation) to maintain their initial volume and no nutrients were added to the systems (to track nutrient decline). Before harvesting (4 weeks after they were transplanted into the system), deionised water was added for the last time (to refill to the initial volume), the solution was allowed to sufficiently mix, and nutrient solution samples were taken again (time T_1_). The plants were then harvested and subjected to biometric measurements and analysis to establish the mineral, pigment, and vitamin C contents.

#### 3.1.5. Microscopic Detection

Microorganism staining and microscopy were performed to verify the presence of microorganisms. Root samples of all variants were fixed with an FAA fixer (formaldehyde, alcohol, acetic acid, distilled water 10%/50%/5%/35%). After fixation, all samples were rinsed three times in deionised water and cleared in 2% KOH for 1 h at 50 °C. The samples were then rinsed four times in deionised H_2_O for 3 min and neutralized in an acid solution (3% HCl) for 5 min at room temperature.

For the observation of *T. harzianum*, samples were prepared according to the approach of Phillips and Hayman [62]. AQP (negative control) and AQP + TricH (sample) variants were immersed in trypan blue solution and incubated for 10 h at room temperature. Subsequently, the samples were rinsed in deionised H_2_O and preserved in lactoglycerol for another 5 h. The samples were then transferred to a glass slide and examined under a VHX-6000 digital microscope (Keyence Corporation, Osaka, Japan).

The HYDRO (negative control), AQP + BM (sample), and suspension of *B. mojavensis* inoculum fixed with 4% paraformaldehyde in phosphate-buffered solution (PBS) (positive control) variants were subjected to fluorescence in situ hybridization (FISH). A positive control was obtained from a bacterial inoculum that was incubated on a shaker for 10 min in PBS with TWEEN; it was centrifuged (10 s, 1000× *g* for perlite separation), cleared with 1 mL of PBS, and centrifuged again (5 min, 11,000× *g* for bacterial cell settlement in pellet form). Bacterial cells were then poured into a 4% paraformaldehyde in PBS, incubated for 15 h at 4 °C, rinsed three times with sterilised deionised H_2_O, and dehydrated in an ethanol series (25, 50, 75, and 96%), each step taking 20 min at 4 °C. Centrifugation (5 min, 11,000× *g*) between the clearing and dehydration steps was performed to separate the cells from the solution. Subsequently, a hybridization solution with Bs-Cy3 and Bs-FAM fluorescent probes (Eurofins Genomics, Ebersberg, Germany) was prepared according to Posada et al. [63]. Samples were transferred to the hybridization solution and incubated in the dark for 2 h at 46 °C. The samples were then immersed in a pre-warmed (46 °C) post-hybridization solution [63] and post-FISH treatment was performed at 48 °C for 20 min. The samples were immediately transferred to a glass slide, covered with VECTASHIELD^®^ HardSet^TM^ Antifade Mounting Medium with DAPI, and observed using a LSM 800, laser scanning microscope, Carl Zeiss AG, Oberkochen, Germany.

#### 3.1.6. Analytical Methodology

The total plant weight, the weight of the above-ground portion (yield), and the root weight were measured, which allowed the calculation of the root/shoot ratio. Total plant weight and root weight were obtained after the subtraction of the average weight of the mineral substrate cube (41 g). The final biometric parameter was the number of leaves per plant. The nutrient solutions were analysed at T_0_ and T_1_ using a colorimetric method (Palintest Photometer 7100, Halma Holdings Ltd., Amersham, UK) in order to determine the level of NO_3_^−^, total ammonia nitrogen (TAN = NH_4_^+^ + NH_3_), PO_4_^3−^, K^+^, Ca^2+^, Mg^2+^, SO_4_^2−^, Na^+^ (calculated from Cl^−^ value), Cl^−^, total Mn (Mn^2+^, Mn^3+^, Mn^7+^, MnO_4_^−^), and F^−^.

The level of potassium, sodium, calcium, and magnesium in the plant samples was determined using isotachophoresis (IONOSEP 2003, Recman–Laboratory Technique Ltd., Ostrava, Czechia), with a leading electrolyte of 7.5 mM H_2_SO_4_ + 7 mM-18-crown-6 + 0.1% Hydroxypropyl MethylCellulose (HPMC1) and a terminating electrolyte of 10 mM BIS-TRIS propane (BTP1). The initial driving current was set to 100 µA and the terminating current to 50 µA. The concentration of phosphorus in the samples was determined through spectrophotometry at a wavelength of 430 nm according to Zbíral et al. [64]. The nitrate levels in the samples were obtained using an ion-selective electrode (type 07–35, Monokrystaly Ltd., Přepeře, Czechia) with a mercury sulphate reference electrode (type RME 121, Monokrystaly Ltd., Přepeře, Czechia) according to Zbíral et al. [64]. The plant pigments (carotenoids, chlorophyll a and b) were extracted in acetone and determined through spectrophotometry at wavelengths of 440, 662, and 644 nm, respectively [65]. Ascorbic acid levels were determined through high performance liquid chromatography (ECOM Ltd., Prague, Czechia) using an ARION^®^ Polar C18 column (5 µm, 150 mm × 4.6 mm) and tetrabutylammonium hydroxide/oxalic acid/water at a ratio of 10/20/70 as the mobile phase. The dry matter content (dry weight, DW) was determined through gravimetry at 105 °C to a constant weight [64].

### 3.2. Biofilter Experiment

When examining the effect of the introduction of the microorganisms under test into the aquaponic system, it was first necessary to test the effect they have on the nitrifying bacteria in the biofilter. For this purpose, samples representing water and plastic segments with nitrifying bacteria were taken from the established biofilter, which is part of the RAS with rainbow trout. The samples were equally divided into three beakers with immediate aeration in order to maintain the bacterial viability. The level of ammonia was checked and adjusted, as needed, using NH_4_Cl to ~6 mg TAN/L in all beakers. The level of TAN, nitrites (NO_2_), and nitrates (NO_3_) were then measured in order to establish their starting levels (T_0_). The beakers were then inoculated. One was inoculated with *T. harzianum* (0.1 g/L of water), the other with *B. mojavensis* (1 g/L of water). The same commercial inoculum used previously was applied. The third beaker was a control that only contained the nitrifying bacteria. TAN, NO_2_, and NO_3_ levels were measured 2 h after inoculation (T_1_) and then after 10 h after inoculation (T_2_). TAN was measured using a colorimetric method (Palintest Photometer 7100, Halma Holdings Ltd., Amersham, UK). The nitrite and nitrate levels were measured using spectrophotometry (VIS Spectrophotometer, Model LI-721, Lasany International, Panchkula, India) at wavelengths of 640 and 420 nm.

### 3.3. Statistical Analysis

All the data were processed using the Statistica 12 software package (DataBon Ltd., Prague, Czechia). Firstly, descriptive statistics were generated for all the data, followed by the Shapiro–Wilk test, which verifies the normality of the data distribution. For data originating from a normal distribution (*p* = 0.05), a multifactorial analysis of variance (ANOVA) was conducted. As part of the post hoc analyses, the homogeneity of the variance was further evaluated using Tukey’s HSD test at a significance level of α = 0.05. The data that did not meet the criteria for a normal distribution were evaluated using the Kruskal–Wallis test, followed by multiple comparisons of mean ranks for all groups (*p* = 0.05).

## 4. Conclusions

PGPMs are considered as an alternative possibility for exploiting the nutrients that are already present in the aquaponic nutrient solution, but in a form that is unavailable to plants. These methods are currently used insufficiently and are being substituted with easier operational methods, such as the addition of mineral fertilizers. The results of the present study demonstrate that *T. harzianum* and *B. mojavensis* can survive in an aquatic environment with a DWC setup and can additionally be beneficial to plants in terms of improved nutrition. In the short term of 10 h, these microorganisms do not have a negative effect on the activity of nitrifying bacteria that are present in RAS biofilter. The results of our study further suggest that the effect of the inoculum in the aquatic environment might be host-specific, indicating the preservation of this phenomenon from the soil environment.

## Figures and Tables

**Figure 1 plants-13-00291-f001:**
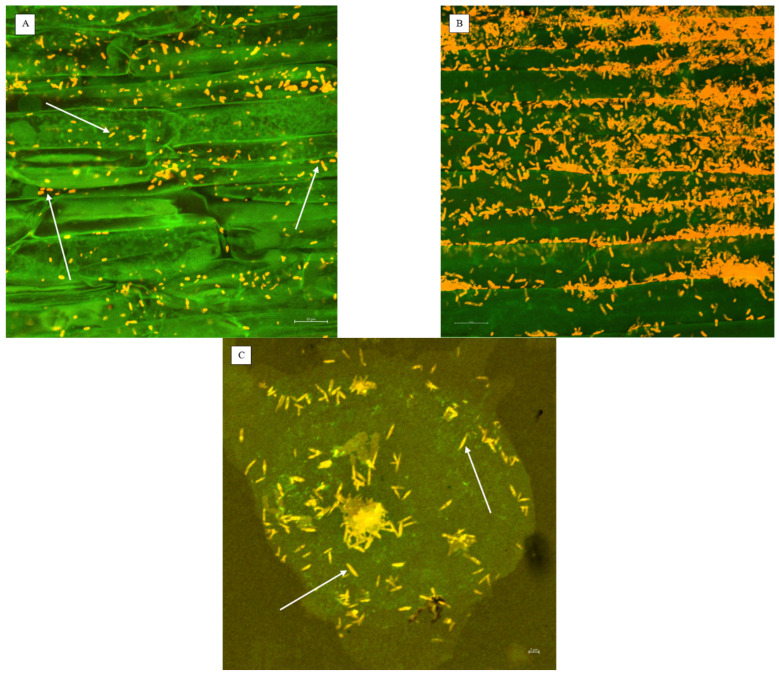
*B. mojavensis* (arrows) detected in root samples of (**A**) basil (10 µm scale), (**B**) ‘Barlach’ lettuce (10 µm scale) that originate from the AQP + BM solution, and (**C**) a positive control (2 µm scale) stained with Bs-Cy3 and Bs-FAM fluorescent probes; refer to Section 3.1.5.

**Figure 2 plants-13-00291-f002:**
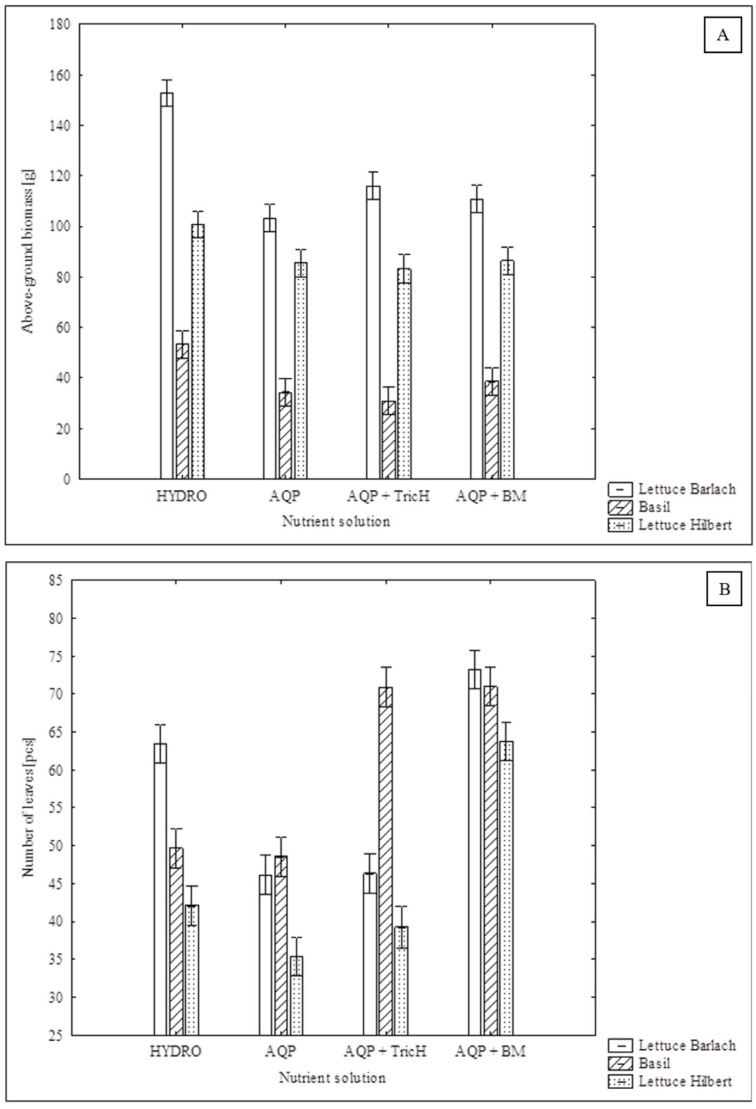
Effect of the nutrient solution on biometric parameters in cultivated crops (**A**) above-ground biomass (yield); (**B**) number of leaves; error bars represent standard error (*n* = 9); the statistical comparison is presented in Table 1.

**Figure 3 plants-13-00291-f003:**
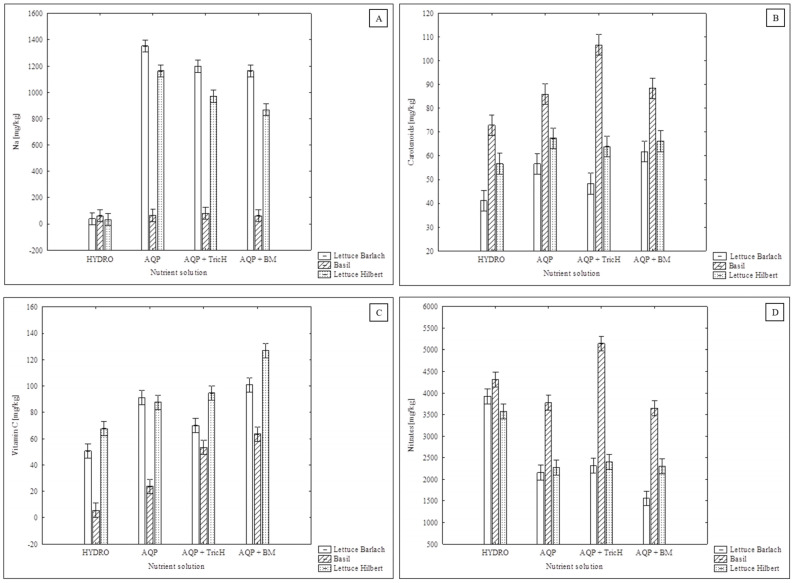
Effect of nutrient solution on quantities of (**A**) sodium, (**B**) carotenoids, (**C**) vitamin C, and (**D**) nitrates in cultivated crops, error bars represent standard error (*n* = 4), the statistical comparison is presented in Table 4.

**Figure 4 plants-13-00291-f004:**
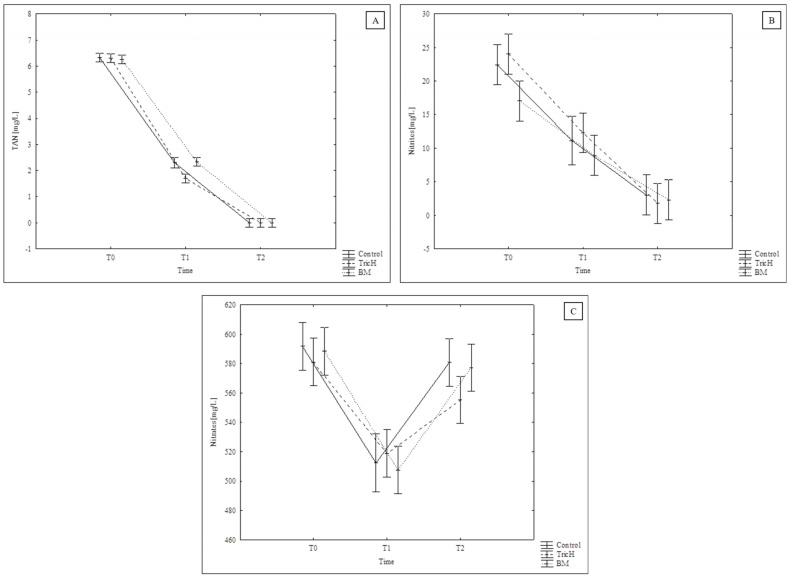
(**A**) Total ammonia N (TAN), (**B**) NO_2_, and (**C**) NO_3_ dynamics during a biofilter experiment (Section 3.2) with a non-inoculated control variant (Control) and variants inoculated with *T. harzianum* (TricH) and *B. mojavensis* (BM); error bars represent standard error (*n* = 3).

**Figure 5 plants-13-00291-f005:**
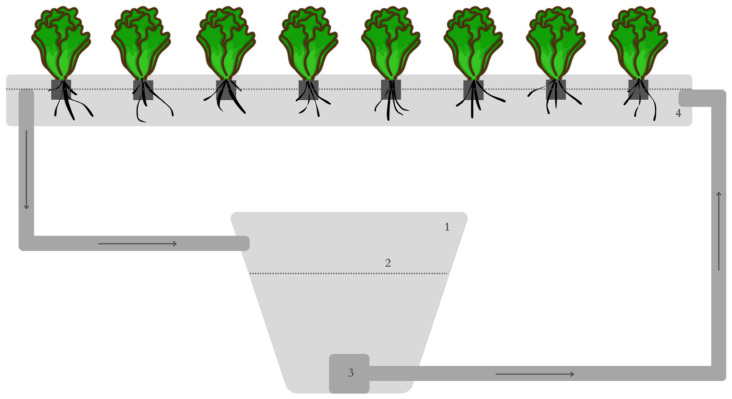
Scheme of a single closed system within the experimental setup: (1) sump tank; (2) nutrient solution level; (3) pump; (4) growing rack.

**Figure 6 plants-13-00291-f006:**
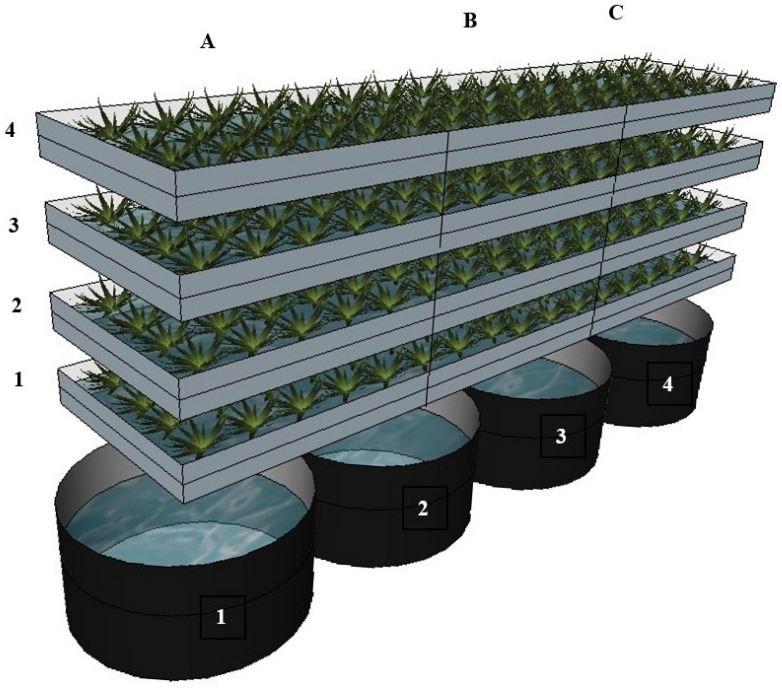
Setup for plant nutrient uptake experiment: (1) hydroponic nutrient solution (HYDRO); (2) aquaponic nutrient solution (AQP); (3) aquaponic nutrient solution inoculated with *Trichoderma harzianum* (AQP + TricH); (4) aquaponic nutrient solution inoculated with *Bacillus mojavensis* (AQP + BM); (A) ‘Hilbert’ lettuce; (B) green basil; (C) ‘Barlach’ lettuce.

**Table 1 plants-13-00291-t001:** Biometric parameters and dry weight of crops in different variants of nutrient solution: the mean value (±standard deviation) for *n* = 9; different lowercase letters in the columns correspond to a significant difference according to Tukey’s HSD test (distribution of homogenous groups, *p* = 0.05); different uppercase letters in the columns correspond to a significant difference according to multiple comparisons of mean ranks for all groups (*p* = 0.05).

Crop	Biometric Parameter	Variant of Nutrient Solution			
		HYDRO	AQP	AQP + TricH	AQP + BM
‘Barlach’ lettuce	Total weight of plants [g]	162.18 (±17.97) A	116.13 (±15.17) B	125.19 (±13.11) AB	120.14 (±10.93) B
	Above-ground biomass [g]	152.78 (±17.00) A	103.32 (±14.50) B	116.00 (±10.68) AB	110.89 (±9.48) B
	Root biomass [g]	9.40 (±1.33) a	12.81 (±1.42) a	9.19 (±2.66) a	9.25 (±1.97) a
	Root:shoot ratio	0.06 (±0.01) a	0.13 (±0.02) a	0.08 (±0.02) a	0.08 (±0.01) a
	Number of leaves [pcs]	63 (±8) a	46 (±8) b	46 (±5) b	73 (±8) a
	Dry weight [%]	4.23 (±0.16) a	5.07 (±0.60) a	4.49 (±0.45) a	4.93 (±0.21) a
Basil	Total weight of plants [g]	69.15 (±18.47) A	48.21 (±8.70) B	43.25 (±5.21) B	52.88 (±10.91) AB
	Above-ground biomass [g]	53.36 (±12.94) A	34.21 (±6.48) B	30.95 (±3.27) B	38.63 (±7.61) AB
	Root biomass [g]	15.78 (±6.00) A	14.00 (±3.09) A	12.29 (±2.36) A	14.26 (±3.83) A
	Root:shoot ratio	0.29 (±0.05) b	0.42 (±0.09) a	0.40 (±0.07) a	0.37 (±0.06) a
	Number of leaves [pcs]	50 (±6) b	49 (±9) b	71 (±7) a	71 (±13) a
	Dry weight [%]	7.30 (±0.89) a	8.44 (±0.71) a	8.63 (±0.96) a	8.50 (±0.69) a
‘Hilbert’ lettuce	Total weight of plants [g]	108.17 (±25.64) a	95.60 (±21.98) a	90.86 (±30.89) a	94.58 (±25.27) a
	Above-ground biomass [g]	100.65 (±23.57) a	85.38 (±18.65) a	83.22 (±28.19) a	86.36 (±22.57) a
	Root biomass [g]	7.51 (±3.31) a	10.21 (±3.59) a	7.65 (±3.03) a	8.23 (±3.58) a
	Root:shoot ratio	0.08 (±0.03) a	0.12 (±0.03) a	0.09 (±0.02) a	0.09 (±0.03) a
	Number of leaves [pcs]	42 (±4) AB	35 (±4) B	39 (±3) B	64 (±11) A
	Dry weight [%]	4.40 (±0.61) a	5.18 (±0.41) a	4.71 (±0.37) a	4.92 (±0.49) a

**Table 2 plants-13-00291-t002:** Differences between the parameters of the nutrient solutions at the beginning (T_0_) and at the end (T_1_) of the experiment: mean value (±standard deviation) (*n* = 3); << the observed value was under the detection limit.

Nutrient Solution	HYDRO		AQP		AQP + TricH		AQP + BM	
Time	T_0_	T_1_	T_0_	T_1_	T_0_	T_1_	T_0_	T_1_
pH	6.10 (±0.00)	6.07 (±0.10)	7.00 (±0.10)	7.17 (±0.06)	7.00 (±0.10)	7.63 (±0.06)	7.00 (±0.10)	7.73 (±0.06)
EC [mS/cm]	1.6 (±0.0)	1.0 (±0.0)	1.4 (±0.0)	1.1 (±0.0)	1.4 (±0.0)	1.2 (±0.0)	1.4 (±0.0)	1.1 (±0.0)
NO_3_^−^ [mg/L]	528 (±14)	317 (±6)	627 (±12)	450 (±10)	627 (±12)	483 (±12)	627 (±12)	420 (±0)
Total ammonia N [mg/L]	6.77 (±0.35)	0.16 (±0.04)	<<	<<	<<	0.16 (±0.02)	<<	0.29 (±0.11)
PO_4_^2−^ [mg/L]	98.3 (±3.0)	97.5 (±2.0)	14.7 (±0.7)	6.2 (±0.1)	14.7 (±0.7)	8.0 (±0.3)	14.7 (±0.7)	8.9 (±0.3)
K^+^ [mg/L]	120.0 (±0.0)	23.7 (±0.6)	19.3 (±0.6)	6.4 (±0.4)	19.3 (±0.6)	6.1 (±0.2)	19.3 (±0.6)	6.1 (±0.2)
Ca^2+^ [mg/L]	45 (±4)	25 (±3)	130 (±5)	88 (±3)	130 (±5)	97 (±3)	130 (±5)	83 (±8)
Mg^2+^ [mg/L]	48 (±1)	35 (±1)	29 (±1)	20 (±1)	29 (±1)	22 (±1)	29 (±1)	21 (±1)
SO4^2−^ [mg/L]	210 (±0)	203 (±3)	119 (±3)	90 (±0)	119 (±3)	98 (±1)	119 (±3)	90 (±4)
Na^+^ [mg/L]	45.8 (±2.9)	38.7 (±1.0)	49.5 (±3.6)	43.6 (±1.4)	49.5 (±3.6)	43.4 (±1.7)	49.5 (±3.6)	29.6 (±1.4)
Cl^−^ [mg/L]	71 (±5)	60 (±2)	76 (±6)	67 (±2)	76 (±6)	67 (±3)	76 (±6)	46 (±2)
Total Mn [mg/L]	2.27 (±0.18)	2.15 (±0.09)	0.28 (±0.07)	0.04 (±0.01)	0.28 (±0.07)	0.13 (±0.08)	0.28 (±0.07)	0.14 (±0.06)
F^−^ [mg/L]	60.72 (±5.45)	46.86 (±5.45)	2.80 (±0.40)	0.68 (±0.32)	2.80 (±0.40)	0.59 (±0.06)	2.80 (±0.40)	0.73 (±0.06)

**Table 3 plants-13-00291-t003:** The parameter dynamics from the beginning of the experiment (T_0_) to the end of the experiment (T_1_): mean value (±standard deviation) for *n* = 3; different lowercase letters in the columns correspond to a significant difference according to Tukey’s HSD test (distribution of homogenous groups, *p* = 0.05); different uppercase letters in the columns correspond to a significant difference according to multiple comparisons of mean ranks for all groups (*p* = 0.05).

Parameter Change T_0_–T_1_	Variant of Nutrient Solution			
	HYDRO	AQP	AQP + TricH	AQP + BM
pH	0.03 (±0.06) a	−0.17 (±0.15) a	−0.63 (±0.15) b	−0.73 (±0.12) b
EC [mS/cm]	0.6 (±0.0) A	0.3 (±0.0) AB	0.2 (±0.0) B	0.3 (±0.0) AB
NO_3_^−^ [mg/L]	211 (±12) a	177 (±15) b	143 (±12) c	207 (±12) ab
Total ammonia N [mg/L]	6.61 (±0.36) A	0.00 (±0.00) AB	−0.16 (±0.02) AB	−0.29 (±0.11) B
PO_4_^2−^ [mg/L]	0.8 (±1.4) c	8.5 (±0.6) a	6.6 (±0.4) ab	5.8 (±0.6) b
K^+^ [mg/L]	96.3 (±0.6) A	12.9 (±0.7) A	13.2 (±0.7) A	13.2 (±0.4) A
Ca^2+^ [mg/L]	21 (±1) c	42 (±6) ab	33 (±6) bc	47 (±6) a
Mg^2+^ [mg/L]	13 (±1) A	9 (±1) A	7 (±1) A	7 (±1) A
SO_4_^2−^ [mg/L]	7 (±3) b	29 (±3) a	21 (±2) a	28 (±6) a
Na^+^ [mg/L]	7.1 (±2.3) b	5.8 (±4.5) b	6.1 (±2.9) b	19.9 (±2.9) a
Cl^−^ [mg/L]	11 (±4) b	9 (±7) b	9 (±5) b	31 (±5) a
Total Mn [mg/L]	0.12 (±0.09) a	0.24 (±0.06) a	0.15 (±0.10) a	0.14 (±0.08) a
F^−^ [mg/L]	13.86 (±7.86) A	2.12 (±0.71) A	2.21 (±0.46) A	2.07 (±0.46) A

**Table 4 plants-13-00291-t004:** Chemical composition of crops in the different variants of nutrient solution: mean value (±standard deviation) for *n* = 4; different lowercase letters in the columns correspond to a significant difference according to Tukey’s HSD test (distribution of homogenous groups, *p* = 0.05); different uppercase letters in the columns correspond to a significant difference according to multiple comparisons of mean ranks for all groups (*p* = 0.05).

Crop	Nutritional Parameter	Variant of Nutrient Solution			
		HYDRO	AQP	AQP + TricH	AQP + BM
‘Barlach’ lettuce	K [mg/kg]	3565.1 (±192.54) A	1220.7 (±212.10) AB	1089.6 (±66.01) B	1174.6 (±143.02) AB
	Na [mg/kg]	39.7 (±6.02) B	1352.5 (±149.04) A	1198.4 (±73.63) AB	1162.7 (±120.01) AB
	Ca [mg/kg]	0.0 (±0.00) B	176.0 (±50.50) AB	418.1 (±78.53) A	167.7 (±252.83) AB
	Mg [mg/kg]	0.0 (±0.00) A	209.7 (±117.56) A	82.5 (±6.93) A	111.5 (±164.50) A
	P [mg/kg]	35.6 (±1.97) A	21.0 (±1.98) AB	20.4 (±1.06) B	21.8 (±0.90) AB
	Chlorophyll a [mg/kg]	162.1 (±5.41) a	185.0 (±33.52) a	183.0 (±32.12) a	203.4 (±14.61) a
	Chlorophyll b [mg/kg]	76.8 (±2.52) a	83.6 (±15.16) a	88.8 (±17.14) a	98.7 (±6.51) a
	Carotenoids [mg/kg]	41.1 (±1.21) a	56.7 (±10.49) a	48.3 (±11.17) a	61.8 (±3.14) a
	Vitamin C [mg/kg]	50.6 (±5.59) d	91.2 (±6.20) bc	70.0 (±2.45) cd	100.9 (±7.89) ab
	Nitrates [mg/kg]	3924.4 (±388.5) A	2156.2 (±195.41) AB	2310.8 (±291.67) AB	1557.6 (±81.38) B
Basil	K [mg/kg]	4647.6 (±640.86) A	1353.9 (±104.40) AB	1458.2 (±219.66) AB	1226.6 (±191.66) B
	Na [mg/kg]	61.0 (±8.42) a	63.9 (±12.26) a	80.8 (±11.26) a	60.8 (±21.56) a
	Ca [mg/kg]	622.2 (±36.54) A	160.3 (±12.59) B	200.1 (±42.66) AB	189.7 (±19.29) AB
	Mg [mg/kg]	131.2 (±15.61) B	819.3 (±61.14) AB	1043.2 (±205.27) A	950.3 (±54.99) A
	P [mg/kg]	92.7 (±17.03) A	33.2 (±2.52) B	45.1 (±7.45) AB	42.8 (±5.56) AB
	Chlorophyll a [mg/kg]	287.4 (±29.93) a	295.0 (±24.08) a	319.1 (±62.20) a	268.4 (±29.10) a
	Chlorophyll b [mg/kg]	125.9 (±9.66) a	126.6 (±11.86) a	135.9 (±29.26) a	112.7 (±14.08) a
	Carotenoids [mg/kg]	73.0 (±9.27) b	86.0 (±5.88) ab	106.8 (±18.33) a	88.5 (±8.84) ab
	Vitamin C [mg/kg]	5.5 (±2.86) b	23.6 (±4.64) b	53.4 (±8.58) a	63.4 (±7.25) a
	Nitrates [mg/kg]	4313.4 (±447.15) ab	3772.8 (±281.35) b	5144.8 (±444.29) a	3651.9 (±391.94) b
‘Hilbert’ lettuce	K [mg/kg]	3606.1 (±435.51) A	1324.3 (±105.02) A	1194.7 (±160.31) A	1253.1 (±257.79) A
	Na [mg/kg]	32.5 (±7.78) B	1162.8 (±195.15) A	970.1 (±118.61) AB	868.4 (±78.98) AB
	Ca [mg/kg]	281.9 (±192.37) A	496.1 (±83.80) A	228.2 (±154.02) A	351.1 (±239.15) A
	Mg [mg/kg]	60.6 (±41.87) A	92.4 (±11.19) A	270.5 (±128.27) A	65.3 (±44.16) A
	P [mg/kg]	36.3 (±5.80) A	21.3 (±1.46) A	21.6 (±2.47) A	22.2 (±2.49) A
	Chlorophyll a [mg/kg]	242.5 (±39.05) a	276.9 (±31.30) a	236.7 (±18.38) a	278.2 (±40.12) a
	Chlorophyll b [mg/kg]	140.3 (±24.58) a	164.3 (±22.58) a	138.5 (±13.43) a	165.1 (±29.00) a
	Carotenoids [mg/kg]	56.7 (±7.93) a	67.4 (±4.54) a	63.9 (±5.03) a	66.2 (±5.95) a
	Vitamin C [mg/kg]	67.6 (±12.88) c	87.5 (±20.45) bc	94.6 (±14.98) b	126.8 (±17.73) a
	Nitrates [mg/kg]	3569.9 (±148.93) a	2269.6 (±524.58) b	2408.0 (±440.33) b	2302.2 (±171.90) b

**Table 5 plants-13-00291-t005:** Phosphorus (PUE), potassium (KUE), calcium (CaUE), and magnesium (MgUE) use efficiency [%]. Calculated as the sum of the analyte fixed in all plants for a particular variant of nutrient solution divided by total input amount of the analyte in the system (180 L), multiplied by 100. Mean value (±standard deviation) for *n* = 3; different uppercase letters in the columns correspond to a significant difference according to multiple comparisons of mean ranks for all groups (*p* = 0.05).

Nutrient Use Efficiency [%]	Variant of Nutrient Solution			
	HYDRO	AQP	AQP + TricH	AQP + BM
PUE	2.19 (±0.07) B	5.37 (±0.24) AB	5.81 (±0.26) AB	6.26 (±0.28) A
KUE	48.15 (±0.00) B	73.88 (±2.17) AB	70.11 (±2.06) AB	73.97 (±2.17) A
CaUE	6.83 (±0.60) A	2.54 (±0.10) AB	2.84 (±0.11) AB	2.17 (±0.08) B
MgUE	1.38 (±0.02) B	10.05 (±0.21) AB	11.23 (±0.23) A	9.55 (±0.20) AB

**Table 6 plants-13-00291-t006:** The dynamics of the total system nitrogen; the calculation of the theoretical nitrogen use efficiency (NUE) [%] value.

	N Input (T_0_)/100 L [g]	N Dynamics (T_0_ − T_1_)/100 L [g]	N Incorporated to Plants in the Form of NO_3_^−^ [g]	N Incorporated in Other Organic Forms/Used by Microbiota [g]	Theoretical NUE [%] in the Case of Utilization of All N from the Solution [%]
HYDRO	12	5	2.4	6.2	40.0
AQP	14	4	1.1	6.1	28.2
AQP + TricH	14	3	1.3	4.6	22.9
AQP + BM	14	5	1.0	7.4	33.0

## Data Availability

The authors declare that all data obtained in this study were reported within the article. Input measured data from MS Excel or Statistica 12 are available upon request from the corresponding author.

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
