# Peer review of "Optimization of Plant Nutrition in Aquaponics: The Impact of Trichoderma harzianum and Bacillus mojavensis on Lettuce and Basil Yield and Mineral Status"

_plants, 2024, doi:10.3390/plants13020291_

Round 1
Reviewer 1 Report
Comments and Suggestions for Authors
Dear authors,
The manuscript entitled "Optimization of Plant Nutrition in Aquaponics: The Impact of Trichoderma harzianum and Bacillus mojavensis on Lettuce and Basil Yield and Mineral Status" presents interesting results. The study was carried out in a correct manner and presents a basis for future studies. It is written in a clear way. I recommend minor modifications before publication:
Line 14-17: in which treatments there was a decrease in these variables observed?
Line 173: Are you sure the unit is CFU/m3? or CFE/mL? please check it
Line 144-147: How much was added to the treatment? What was the final solution concentration? This nutrient solution does not have all the essential elements, how were the requirements of all the essential elements for optimal growth met in your control group?
Line 286: Are you sure it is "too high"? What concentration is too high to prevent the growth of these plant species? This part needs to be discussed. Many glycophyte species grow without major problems at concentrations of 1000 mg/L NaCl, halophyte plants grow at concentrations higher than 11 g/L (200 mM NaCl), here some examples: https://www.sciencedirect.com/science/article/pii/S0098847213000452, https://academic.oup.com/aobpla/article/doi/10.1093/aobpla/plv020/200235
Paragraph 3.1.1: is it necessary to discuss why lettuce shows significant differences but the other species do not, higher nutritional requirements?
Fig. 2: Error bars represent standard deviation or standard error? of how many replicates?
Line 349: 14,2-1,6: change comma to decimal point
paragraph 3.1.3: How can you be sure that in HYDRO, precipitation of P and Ca occurred? It should not occur at these concentrations in hydroponics
paragraph 3.1.3: "...suggest the incorporation of halophytes in the hydroponic part of the aquaponic system, but this needs further study in the future to establish the species and quantity of plants that might be needed to prevent accumulation.", this is not really true, halophytes are very well studied, also in hydroponics/aquaponics (https://www.sciencedirect.com/science/article/pii/S0044848620312928, https://www.mdpi.com/2071-1050/6/2/836 etc)
Any explanation for the difference in density of microorganisms in figure 5?
All the best
Reviewer 2 Report
Comments and Suggestions for Authors
Optimization of Plant Nutrition in Aquaponics: The Impact of Trichoderma harzianum and Bacillus mojavensis on Lettuce and Basil Yield and Mineral Status
In current study, the authors aimed to test the effect of a nutrient solution, with the addition of microbial inoculum, on the growth and mineral composition of 'Hilbert' and 'Barlach' lettuce cultivars (Lactuca sativa var. crispa, L.) and basil (Ocimum basilicum, L.) cultivated in a vertical indoor farm. The results showed that the effect of T. harzianum inoculation was particularly evident in basil, with an increase in the number of leaves, the content of nitrates and vitamin C, while there was no change in yield recorded when compared to the non-inoculated AQP variant. Therefore, T. harzianum inoculation may be recommended for growing basil in N-limited conditions. Inoculation with B. mojavesis resulted in a significantly higher uptake of Na+ and Cl- ions from the solution while simultaneously producing a lower sodium content in the 'Hilbert' lettuce. B. mojavensis significantly increased PUE and the number of leaves in all the crops examined and for the basil and 'Hilbert' lettuce there was a significant increase in the vitamin C content with no effect on yield. Inoculation with B. mojavensis can therefore be recommended for aquaponic cultivation, as it partially mitigates the accumulation of Na+ and Cl- in the solution while simultaneously it has a positive impact on certain aspects of the crop. These findings can help to reduce the required level of supplemental mineral fertilizers in aquaponics.
The MS is clearly written but some issues should be considered to improve the quality of the manuscript. The comments and suggestions are as follows:
- Abstract: The abstract is a little big long. I recommend authors revise it carefully and be more concise. The importance of the study is missed in the abstract as well.
- Line 80. Inoculation should be inoculation.
- Line 82. phyt-ohormones should be phyto-hormones
- Lines 108-112. I think there is no need for this part. What is the relation between this part and the current study?
- Line 96. Before (Their ability …..) I recommend authors to add: The effective role of PGPR in enhancing the plant growth, yield and quality has been reported (Kahil et al., 2017). Otherwise, the production of chemical-free products from edible crops has gained more attention to guarantee their quality and safety (Hassan et al., 2012).
AA Kahil, FAS Hassan, EF Ali. Influence of bio-fertilizers on growth, yield and anthocyanin content of Hibiscus sabdariffa L. plant under Taif region conditions. Annual Research and Review in Biology 17, 2017. 10.9734/ARRB/2017/36099
Hassan, F.; Ali, E.F.; Mahfouz, S. Comparison between different fertilization sources, irrigation frequency and their combinations on the growth and yield of the coriander plant. Aust. J. Appl. Basic Sci. 2012, 6, 600-615.
- Line 97-106. This paragraph is well written and clearly summarized the main mode of actions to promote plant growth by PGPR. However, by the end of this paragraph the authors limit these effects under stress conditions. Therefore, I recommend authors to modify this sentence.
- The objective of the study, I think, need revision in the light of the paragraph presented in Lines 60-73. In my opinion, the authors did not aim, only, to assess how lettuce and basil, cultivated in nutrient solutions derived from various nutrient sources as they mentioned.
- Line 140. Although the System setup and Experimental design are clear, the authors did not mention the type of design. Is it CRD, split or factorial? How the authors arrange the treatments? Is this experiment repeated two or three times or only one time?
- Line 144. Is HYDRO treatment set as a control?
- Lines 170 and 174. Why the authors selected one concentration only from each of Trichoderma or Bacillus? What is the basis for selecting this concentration?
- Line 197. Microscopic detection. There is a confusion in considering the control. In T. harzianum, AQP is considered a negative control. While in B. mojavensis The HYDRO is considered a negative control. Why?
- In Fig. 2. and Fig. 3. Please explain in the legend what is bar above the column refer to? Also the small letters above the columns.
- Conclusions
- The conclusion is long. Please carefully revise it and put the final recommendations based on your study. Please focus only on the message that you want to deliver.
Comments on the Quality of English Language
Minor editing of English language is required.
Reviewer 3 Report
Comments and Suggestions for Authors
This manuscript systematically studied the effects of aquaponics with the addition of Trichoderma and Bacillus on the yield and mineral status of lettuce and basil. The manuscript provides some information of interest to the reader.
Minor points:
1) Abstract is too long, please summarize well and rewrite the abstract;
2) Please rewrite the ‘Keywords’: what is PGPMs? And why authors choose vitamin C as the key word?
3) Line82, ‘phyot-ohormones’ should be ‘phyto-hormones’;
4) Line145, ‘0,4%’ described as ‘0.4%’ may be a good choice, please check and revise them throughout the manuscript;
5) The contents of Table 1 contain the Figure 2’ content, please revise;
6) Figure 4 and Figure 5 may look better horizontally than vertically.
Round 2
Reviewer 2 Report
Comments and Suggestions for Authors
The authors responded to all comments. Actually, This version is significantly improved and I accept it.
Author Response
Once again, thank you very much for the revisions and accepting the manuscript.
With best regards,
authors